# Evaluation of Human Mesenchymal Stromal Cells as Carriers for the Delivery of Oncolytic HAdV-5 to Head and Neck Squamous Cell Carcinomas

**DOI:** 10.3390/v15010218

**Published:** 2023-01-13

**Authors:** Robin Nilson, Lea Krutzke, Frederik Wienen, Markus Rojewski, Philip Helge Zeplin, Wolfgang Funk, Hubert Schrezenmeier, Stefan Kochanek, Astrid Kritzinger

**Affiliations:** 1Department of Gene Therapy, University Medical Center Ulm, 89081 Ulm, Germany; 2Institute for Transfusion Medicine, University Medical Center Ulm, 89081 Ulm, Germany; 3Institute for Clinical Transfusion Medicine and Immunogenetics Ulm, German Red Cross Blood Donation Service, 89081 Ulm, Germany; 4Schlosspark Klinik Ludwigsburg, Privatklinik für Plastische und Ästhetische Chirurgie, 71638 Ludwigsburg, Germany; 5Schönheitsklinik Dr. Funk, 81739 München, Germany

**Keywords:** human mesenchymal stromal cells, mesenchymal stem cells, carrier cells, human head and neck squamous cell carcinoma, migration, oncolytic adenovirus

## Abstract

Human multipotent mesenchymal stromal cells (hMSCs) are of significant therapeutic interest due to their ability to deliver oncolytic adenoviruses to tumors. This approach is also investigated for targeting head and neck squamous cell carcinomas (HNSCCs). HAdV-5-HexPos3, a recently reported capsid-modified vector based on human adenovirus type 5 (HAdV-5), showed strongly improved infection of both hMSCs and the HNSCC cell line UM-SCC-11B. Given that, we generated life cycle-unmodified and -modified replication-competent HAdV-5-HexPos3 vector variants and analyzed their replication within bone marrow- and adipose tissue-derived hMSCs. Efficient replication was detected for both life cycle-unmodified and -modified vectors. Moreover, we analyzed the migration of vector-carrying hMSCs toward different HNSCCs. Although migration of hMSCs to HNSCC cell lines was confirmed in vitro, no homing of hMSCs to HNSCC xenografts was observed in vivo in mice and in ovo in a chorioallantoic membrane model. Taken together, our data suggest that HAdV-5-HexPos3 is a potent candidate for hMSC-based oncolytic therapy of HNSCCs. However, it also emphasizes the importance of generating optimized in vivo models for the evaluation of hMSC as carrier cells.

## 1. Introduction

In 2020, more than 250,000 people died from head and neck squamous cell carcinomas (HNSCCs), a cancer type derived from the mucosal epithelium of the larynx, the pharynx, or the oral cavity [1,2]. Despite surgical advances and aggressive multimodality therapy, the prognosis for advanced HNSCCs is poor (<25% survival rate after five years) [2,3]. One promising approach to improve the treatment of HNSCCs is oncolytic virotherapy, which aims to destroy tumors by direct virus-mediated cell lysis and by the induction of antitumor immunity [4], as it is actively pursued with oncolytic adenoviral vectors (oAVs) [5,6,7].

Although oAVs may harbor significant potential for the lysis of tumor cells in vivo, clinical efficacy is limited for several reasons. One reason is an unfavorable in vivo biodistribution pattern of oAVs: upon systemic administration, the viral particles encounter several barriers that lead to extensive off-target sequestration, including binding to human erythrocytes [8,9], neutralization by natural IgMs [10,11], complement proteins [12], and pre-existing IgGs [13], transduction of hepatocytes [14,15], and scavenger receptor-mediated sequestration by macrophages [16,17,18]. Although off-target sequestration can be reduced by intratumoral injection of oAVs, this route of application limits the therapeutic efficacy due to the restricted spread of viral particles within the tumor tissue and the inability to reach metastases [19]. Hence, a strategy enabling the systemic administration of oAVs without off-target sequestration might improve the efficiency of oncolytic virotherapy.

One strategy is based on the use of human mesenchymal stromal cells (hMSCs) as carriers to transport viral particles to the tumor site while shielding them from blood components. In this approach, hMSCs are infected with oAVs ex vivo and subsequently administered systemically to a patient. Attracted by chemokines and growth factors released from the tumor, hMSCs are believed to infiltrate the tumor site and are expected to be lysed by the replicating oAVs in direct vicinity of the targeted tumor cells. Consequently, the released viral progeny can infect and lyse tumor cells, ideally leading to tumor necrosis and the exposition of cancer-associated antigens, which can initiate the reactivation of the immune system [20]. This approach showed encouraging results in many preclinical studies and is already being evaluated in the clinic (e.g., NCT03896568, NCT02068794, NCT01844661) [21,22].

As hMSCs lack expression of the coxsackievirus and adenovirus receptor (CAR), they are hardly infected by HAdV-5 vectors, which is why capsid-modified variants have been developed to enable more efficient infection [23]. Frequently used examples are RGD-modified HAdV-5 vectors (targeting αvβ3 and αvβ5 integrins) [24,25], or chimeric HAdV-5/3 vectors (targeting desmoglein-2 and CD46) [26,27,28]. Recently, we reported a novel HAdV-5 mutant vector—HAdV-5-HexPos3—which carries a modified hexon protein: 13 (mostly negatively charged) amino acids were deleted and replaced by four consecutive positively charged lysine residues [29]. This modification enabled highly efficient CAR-independent infection of hMSCs and many other target cells, including the HNSCC cell line UM-SCC-11B [29]. Thus, we hypothesized that HAdV-5-HexPos3 vectors might be well suited for hMSC-based oncolytic virotherapy to treat HNSCCs.

The overall aim of our study was to evaluate the therapeutic potential of hMSCs loaded with replication-competent HAdV-5-HexPos3 vectors for treating HNSCCs in a preclinical model. We generated several replication-competent HAdV-5-HexPos3 vectors harboring different life cycle modifications to assess their replication behavior within bone marrow- (BM) and adipose tissue-derived (A-) hMSCs. Moreover, we analyzed the migration of non-infected and infected hMSCs toward different HNSCC cell lines. Although we observed robust migration toward these tumor cells in vitro, no migration of vector-loaded hMSCs to HNSCC xenograft tumors was detected in two mouse models and a chorioallantoic membrane (CAM) model. Our results provide evidence that replication-competent HAdV-5-HexPos3 vectors might be a valuable tool for hMSC-mediated oncolytic virotherapy and highlight the need to improve site-directed homing of hMSCs toward tumors.

## 2. Materials and Methods

### 2.1. Cell Lines

All cell lines were cultured adherently on culture dishes (N52.E6: ThermoFisher Scientific, Waltham, MA, United States, #168381; other cell lines: Sarstedt, Nümbrecht, Germany, #83.3903;) at 37 °C, 90% relative humidity, and 5% CO_2_. Cells were detached using Trypsin-EDTA 0.05% (ThermoFisher Scientific, #25300054). All culture media were supplemented with 10% FBS (ThermoFisher Scientific, #10270-106) and 1x Penicillin/Streptomycin/Glutamine (ThermoFisher Scientific, #10378-016).

All HNSCC cell lines (UD-SCC-1 (Expasy #CVCL_E324), UD-SCC-2 (Expasy #CVCL_E325), UD-SCC-5 (Expasy #CVCL_L548), UD-SCC-6 (Expasy #CVCL_M120), UD-SCC-8 (Expasy #CVCL_YD74), UM-SCC-10A (Expasy #CVCL_7713), UM-SCC-11B (Expasy #CVCL_7716), UM-SCC-17B (Expasy #CVCL_7725), UT-SCC-24A (Expasy #CVCL_7826), and UT-SCC-50 (Expasy #CVCL_7859)) were cultured in DMEM (ThermoFisher Scientific, #41966-029).

A549 cells (ATCC #CCL-185) were cultivated in MEM (ThermoFisher Scientific, #31095-029).

N52.E6 cells ([30]) were cultivated in α-MEM (ThermoFisher Scientific, #22561-021).

### 2.2. Human MSC Isolation, Characterization, and Cultivation

BM- and A-hMSCs were isolated and characterized following high-quality standards as described previously [31,32]. Briefly, the characterization of BM-hMSCs comprised analysis of expression (CD105, CD73, and CD90) or absence (CD45, CD34, CD3, and HLA-DQ, -DP, -DR) of specific surface markers, plastic-adherent growth, and trilineage differentiation following the ISCT minimal criteria [33]. A-hMSCs were analyzed for expression (CD13, CD105, CD73, CD90) and absence (CD45, CD34, CD14, HLA-DQ, -DP, -DR) of specific surface markers based on the IFATS and the ISCT joint statement [33]. Human MSCs were propagated in BioWhittaker Alpha Minimum Essential Medium (Lonza, Basel, Switzerland, #BE02-002F) supplemented with 5% (A-hMSCs) or 8% (BM-hMSCs) irradiated human platelet lysate and 500 units of heparin (Ratiopharm, Ulm, Germany) [34]. Human MSCs were cultivated adherently on cell culture plates (ThermoFisher Scientific, #168381) at 37 °C, 90% relative humidity, and 5% CO_2_. TrypLE select (ThermoFisher Scientific, #12563029) was used to detach hMSCs from the plates. Human MSCs were used for experiments up to passage 8. All experiments were performed with hMSCs isolated from three healthy donors.

### 2.3. Adenoviral Vectors

All adenoviral vectors used in this study are based on HAdV-5 (GenBank AY339865.1). Replication-incompetent first-generation vectors carried a CMV promoter-driven enhanced GFP (eGFP) or firefly luciferase-eGFP fusion protein cassette (eGFP derived from pEGFP-N1 plasmid, Takara Bio, San Jose, CA, USA, #6085-1; firefly luciferase GeneBank ID: MK484107.1, nucleotides (nt) 283 to 1932) inserted in reverse orientation at the position of the deleted *E1* (Δ nt 441-3,522). They were produced in N52.E6 cells [30].

All replication-competent vectors carried an CMV promoter-driven eGFP-Nano luciferase expression cassette (eGFP derived from pEGFP-N1 plasmid, Takara Bio, #6085-1, Nano luciferase derived from pNL1.1. plasmid, Promega, Madison, WI, USA, #N109A) inserted between the *E1A* and *E1B* genes (GenBank ID: AY339865.1 nt 1648/1649). They were produced in A549 cells.

E1A-Δ24 vectors had a deletion of 24 base pairs (bp) in the *E1A 12S* gene (GenBank ID: AY339865.1 nt 919 to 943) [35].

E1B-Δ19K vectors had a deletion of 147 bp in the *E1B* gene (GenBank ID: AY339865.1 nt 1770 to 1916) [36].

HexPos3 vectors have 13 aa deleted in Hexon HVR1 and replaced by four lysines (EEEDDDNEDEVDE → KKKK; GenBank ID: AY339865.1 nt 19280-19318) [29].

ΔCAR vectors had a point mutation in the fiber knob (Y477A), which significantly reduces binding of the HAdV-5 fiber knob to CAR [37].

Production, CsCl gradient-based purification, titer determination, and quality control of HAdV-5 vectors have been described previously [23,29]. Briefly, cells (N52.E6 for production of replication-incompetent first-generation vectors, A549 for production of replication-competent vectors) were harvested when the cytopathic effect (CPE) was fully developed (usually ~48 h post-infection), centrifuged at 300× *g* for 10 min, and the supernatant was discarded. The cell pellet was resuspended in 3 mL HEPES buffer (50 mM HEPES and 150 mM (wildtype capsid vectors) or 250 mM (HAdV-5-HexPos3 vectors) NaCl, pH 7.4) per gradient, equaling ten 15 cm culture dishes, and cells were lysed by three freeze-thaw cycles. Cell debris was removed by centrifugation, and HAdV-5 particles were purified from the lysate using two consecutive CsCl-gradients (step gradient: 3 mL CsCl with ρ = 1.41 g/mL, overlayed by 5 mL CsCl with ρ = 1.27 g/mL and 3 mL lysate, centrifugation for 2 h at 176,000× *g*, 4 °C; continuous gradient: 11.5 mL CsCl with ρ = 1.37 g/mL mixed with 2.5 mL of viral vector from the step gradient, centrifugation for 20 h at 176,000× *g*, 4 °C; CsCl dissolved in 50 mM HEPES, 150 mM NaCl, pH 7.4; centrifugation was performed using Ultraclear centrifuge tubes (Beckman Coulter, Brea, CA, USA, #344059) and a Sorvall Discovery 90SE Ultracentrifuge with TH-641 rotor). Aspirated viral particles were desalted using PD-10 size exclusion columns (Cytiva, Marlborough, MA, USA, #17085101) and stored at −80 °C (storage buffer: 50 mM HEPES, 150 mM NaCl, pH 7.4 with 10% glycerol).

Total physical/particle titers were determined by optical density measurement at 260 nm wavelength [38]. SDS-PAGE with subsequent silver staining [39] was performed to confirm the purity of viral vectors. Moreover, vector genome integrity was verified by restriction enzyme digestion and sequencing of isolated vector DNA.

### 2.4. Transduction/Infection of hMSCs with HAdV-5 Vectors

Transduction or infection of hMSCs was performed in 24- (3 × 10^4^ cells/well, 1 mL medium, ThermoFisher Scientific #142485) or 6- (1 × 10^5^ cells/well, 3 mL medium, ThermoFisher Scientific #40685) well plates. One day after seeding, cells were washed with PBS, and transduced/infected with the indicated HAdV-5 vector and physical/particle multiplicity of infection (pMOI) in supplement-free medium. In the case of transduction using human blood coagulation factor X (hFX) as a transduction enhancer, viral particles were pre-incubated with 4 fg of hFX/viral particle (30 min, 37 °C.) Three hours post-transduction, the vector-containing medium was removed, cells were washed with PBS, and fresh medium was added. For in vivo, in ovo, and in vitro migration analysis, hMSCs were washed with PBS 3 h post-infection. Subsequently, cells were detached from the plates using TrypLE select (ThermoFisher Scientific, #12563029), counted with a Neubauer counting chamber, and diluted to the respective cell concentration in culture medium (in vitro experiments) or PBS (in vivo and in ovo experiments).

### 2.5. HNSCC Xenograft Mouse Model

All animal experiments were conducted in an Association for Assessment and Accreditation of Laboratory Animal Care International-accredited facility in accordance with the European Union (EU) Directive 2010/63/EU for animal experiments. Authorization was granted by the Animal Care Commission of the government of Baden Württemberg, Germany (Regierungspräsidium Tübingen, TVA #1459).

2 × 10^6^ UD-SCC-2 or UD-SCC-6 cells were diluted in 50 µL PBS, mixed with 50 µL growth factor-reduced matrigel (Merck, Burlington, MA, USA, #CLS356252), and injected subcutaneously into the left flank of NSG mice (bred in the animal research center Ulm). Subsequently, mice were monitored daily (general health status, body weight, tumor volume). Using a digital caliper, the tumor volume was determined by measuring the greatest longitudinal diameter (length) and the greatest transverse diameter (width). Subsequently, the tumor volume was calculated by the modified ellipsoidal formula: tumor volume = 0.5 × length × width^2^.

### 2.6. Analysis of In Vivo hMSC Biodistribution Using the IVIS 200 System

After 14 or 21 days of xenograft tumor growth in NSG mice, 1 × 10^6^ BM-hMSCs (non-transduced or transduced with an eGFP-firefly luciferase-expressing replication-incompetent vector with pMOI 1000 3 h before injection) diluted in 100 µL PBS were injected intravenously (i.v.) into the tail vein or intraperitoneally (i.p.). Luciferase activity was tracked in the IVIS 200 system (Caliper Life Sciences, Hopkinton, MA, USA) upon injection of the luciferase substrate (VivoGlo^®^ luciferin, Promega, #P1041, 10 mg/mL in PBS, 300 µL injected i.p) at indicated time points.

### 2.7. Tissue Preparation for Immunohistochemistry

At the end of the in vivo NSG mice experiments, mice were sacrificed, and organs and tumors were harvested and frozen in liquid nitrogen. Moreover, tumor tissue samples were fixed (2% PFA in PBS, 4 °C, 24 h) and incubated in cryoprotectant (20% sucrose, 4 °C, 24 h). Subsequently, samples were embedded in TissueTek (Sakura Finetek, Staufen im Breisgau, Germany, #4583) and frozen at −80 °C before the preparation of cryosections. Cryosections (6 μm) of tumor tissue were prepared in a cryostat cryotome (Leica, Wetzlar, Germany), transferred to microscope slides (ThermoFisher Scientific, #10149870), and used for immunohistochemical staining.

### 2.8. Analysis of hMSC Migration in the Chorioallantoic Membrane (CAM) Model

UM-SCC-11B cells were engrafted on the CAM of fertilized chicken eggs as described in detail previously [40]. Four days after seeding UM-SCC-11B cells on the CAM, 5 × 10^5^ BM-hMSCs (transduced with an eGFP-expressing replication-incompetent HAdV-5-HexPos3 vector with pMOI 1000 3 h before injection) or 5 × 10^9^ pure eGFP-expressing replication-incompetent HAdV-5-HexPos3 particles were injected i.v. (both dissolved in 50 µL PBS). Moreover, transduced hMSCs were also injected intratumorally (i.t.). A total of 48 h post-injection, chicks were sacrificed as described previously [40], and tumors were harvested and either frozen in liquid nitrogen for subsequent qPCR analysis or fixed (2% PFA in PBS, 4 °C, 24 h) and incubated in cryoprotectant (20% sucrose in PBS, 4 °C, 24 h) before preparing 6 µm cryosections. Cryosections were analyzed for eGFP expression using fluorescence microscopy.

### 2.9. Immunohistochemical Staining of CD31

Cryosections were air-dried, followed by antigen-retrieval (30 s boiling in 10 mM Citric Acid, 0.05% Tween 20, pH 6.0). Subsequently, sections were permeabilized (0.5% Triton X in PBS, 30 min) and blocked (5% BSA and 10% goat serum in PBS, 1 h). After washing (PBS with 0.05% Tween20), sections were incubated with both rat α-mouse CD31 antibody (1:200, BD Biosciences #557355) and mouse α-human nuclear antigen antibody (1:100, Abcam #ab191181) in 250 µL antibody diluent (Agilent, Santa Clara, CA, USA, #S0809) (4 °C, overnight). After washing, sections were incubated with the respective secondary antibodies (Alexa fluor 488-labeled goat α-rat, Invitrogen #A-11006, and Alexa fluor 594-labeled goat α-mouse, Invitrogen #A-11032, both diluted 1:10.000, 1 h, 4 °C). Next, all cell nuclei were stained with 4′,6-diamidino-2-phenylindole (DAPI, Sigma Aldrich, #D9542-1MG; 200 µL, 100 ng/mL in PBS, RT, 1 min). After washing, sections were covered with a fluorescence mounting medium (Dako, #53023). As a control, staining with an isotype control (Jackson Immunoresearch Laboratories, Westgrove, PA, USA, #012-000-003) and secondary antibody only was performed.

### 2.10. DNA Extraction from Tissue Samples and Analysis of Adenoviral E4 Copy Number Using Quantitative Real-Time Polymerase Chain Reaction (qPCR)

In vivo biodistribution of hMSC was analyzed by the detection of the adenoviral genome copy number in tumors and organs. First, DNA was extracted from tumors and organs using the GenElute Mammalian Genomic DNA Miniprep Kit (Sigma Aldrich, St. Louis, MO, USA, #G1N350-1KT) following the manufacturer’s instructions. After eluting the DNA in 10 mM Tris, pH 8.0, DNA concentration was determined photometrically at 260 nm. To determine the number of adenoviral genomes in the samples, qPCR analysis targeting the adenoviral *E4* gene was performed using KAPA SYBR^®^ FAST (Kapa Biosystems, Wilmington, MA, USA#KK4502) according to the manufacturer’s instructions and a CFX Connect qPCR Cycler (BioRad, Hercules, CA, USA). As a template, 20 ng of total DNA (dissolved in 2 µL 10 mM Tris, pH 8.0) was used. Detected adenoviral genome copy numbers were normalized to detected *actin* copy numbers. Primers specific for the adenoviral *E4* region: forw.: 5′ TAGACGATCCCTACTGTACG 3′; rev.: 5′ GGAAATATGACTACGTCCGG 3′. Primers specific for β-actin: forw.: 5′ GCTCCTCCTGAGCGCAAG 3′; rev.: 5′ CATCTGCTGGAAGGTGGACA 3′.

### 2.11. Virus Replication Assay

3 × 10^4^ hMSCs were seeded in 24-well plates (Thermo Fisher Scientific, #142485). The next day, hMSCs were infected with indicated pMOIs. Six days post-infection, hMSCs were washed and fixed with 4% PFA in PBS (15 min, RT). After washing with PBS, hMSCs were stained with 0.1% crystal violet dissolved in H_2_O (2 min, RT). After three washing steps with PBS, the stained plate was air-dried for some minutes. Subsequently, pictures of the plates were taken.

### 2.12. Quantification of Virus Replication in hMSCs by qPCR

Human MSCs were infected with the indicated replication-competent HAdV-5-HexPos3 vectors. At the indicated time points, supernatants and cells (detached with a cell scraper) were harvested, adjusted to a volume of 2 mL, and snap-frozen. Viral particles were released from the cells by three freeze-thaw cycles. To digest all non-encapsidated adenoviral genomes, 25 µL of the samples were transferred to new tubes and treated with 5 units of DNAse I (37 °C for 30 min). DNAse I and the viral particles were denatured at 95 °C for 10 min. Subsequently, 2 µL of the DNAse I-treated samples were used for qPCR analysis to quantify *E4* copy numbers (described in 2.10). Subsequently, based on the *E4* copy numbers detected in 2 µL of sample, the total amount of *E4* copies in the hMSC lysate (total volume 2000 µL) was calculated and divided by the total number of cells per well (3 × 10^4^ hMSCs) to determine the total amount of genome-containing adenoviral particles produced per hMSC.

### 2.13. Analysis of Lactate Dehydrogenase (LDH) Release to Cell Culture Supernatant

The amount of LDH released to the cell culture supernatant of hMSC cultures upon HAdV-5 infection was determined using the LDH-Glo™ Cytotoxicity Assay (Promega, #J2380) following the manufacturer’s instructions. Briefly, hMSCs were infected with pMOI 900 of the indicated replication-competent HAdV-5-HexPos3 vectors. Starting 24 h post-infection, 50 µL of supernatant were collected daily and centrifuged (400× *g*, 10 min). A total of 20 µL of the supernatant were diluted in 480 µL of LDH Assay storage buffer and used for analysis. Luminescence signals were quantified using the GloMax^®^ luminometer (Promega).

### 2.14. Analysis of Virus Spread by Plaque Assays

Two hundred hMSCs were infected with the indicated HAdV-5 vectors (pMOI 900, in supplement-free medium) and seeded on top of UD-SCC-2 cells (seeded the day before in 6-well plates, 1 × 10^6^ cells/well). The next day, the medium was removed, and cells were overlayed with 3 mL of 0.75% low-melting agarose (Biozym, Hessisch Oldendorf, Germany, #840101) solved in MEM medium (ThermoFisher Scientific, #31095-029) supplemented with 10% FBS. Subsequently, cells were incubated at 37 °C and analyzed for the spread of eGFP signal by fluorescence microscopy daily until 216 h post-infection. The percentage of non-lysed BM-hMSCs was determined by setting the number of single cells (=non-lysed cells) and the number of eGFP-positive plaques in proportion. Moreover, the diameter of the occurring plaques was determined.

### 2.15. Migration Assays In Vitro

For Boyden chamber migration assays, 2 × 10^4^ hMSCs (non-infected or 3 h post-infection with pMOI 900) were seeded into the upper chamber of tissue culture (TC)-inserts with a pore size of 8 µm. The TC-inserts were previously placed into 24-well plates harboring either (i) tumor cell-conditioned medium (harvested from tumor cell lines UD-SCC-2 or UD-SCC-6 cultured on 15 cm dishes for three-four days in DMEM without any additives) or (ii) the respective culture medium (DMEM without any additives) as a negative control.

After 18 h, non-migrated hMSCs were removed by thoroughly cleaning the upper chamber with cotton swabs. Subsequently, migrated cells at the bottom side of the TC insert were fixed with 4% PFA in PBS (15 min, RT). Cell nuclei were stained with 4′,6-diamidino-2-phenylindole (DAPI, Sigma Aldrich, #D9542-1MG; 1 µg/mL) to allow for the counting of migrated hMSCs. Subsequently, three images of different areas of the TC-inserts were taken using a fluorescence microscope (10-fold magnification). The total number of migrated hMSCs per TC insert was calculated based on the average number of cell nuclei counted per microscope image.

### 2.16. Statistical Analysis

Experiments performed in this study were repeated at least three times independently unless described otherwise. Statistical tests used for analysis are stated in the figure legends and were performed using GraphPad Prism software version 8.4.2 (GraphPad Software LLC, San Diego, CA, USA).

## 3. Results

### 3.1. Replication-Competent HAdV-5-HexPos3 Particles Efficiently Replicate in BM- and A-hMSCs

Modification of the HAdV-5 hexon protein by deleting a stretch of 13 amino acids and replacing them with four consecutive lysine residues was previously shown to result in very efficient infection of hMSCs and the HNSCC cell line UM-SCC-11B (for both ≥80% eGFP-positive cells with pMOI 1000) [29]. Thus, we hypothesized that combining oncolytic HAdV-5-HexPos3 vectors with hMSCs as carrier cells might be a promising approach for the treatment of HNSCCs. We generated three different types of replication-competent vectors: (i) vectors harboring the complete wildtype HAdV-5 genome (referred to as ‘no life cycle modification’), (ii) HAdV-5-E1A-Δ24 vectors harboring a 24 base pair (bp) deletion in the *E1A 12S* gene, which is supposed to restrict virus replication to proliferating cells, and (iii) HAdV-5-E1A-Δ24-E1B-Δ19K vectors, which carry, additionally to the E1A 12S modification, a deletion of the *E1B 19K* gene sequence, supposed to restrict virus replication to cells with mutated apoptotic pathways.

To analyze the ability of the three vector types to replicate in hMSCs, bone marrow- (BM-) and adipose tissue-derived (A-) hMSCs were infected. In addition to the infection with replication-competent HAdV-5-HexPos3 particles, hMSCs were also infected with their wildtype-capsid counterparts (both pMOI 1000). To overcome the inefficient infection of hMSCs with wildtype-capsid HAdV-5 particles, those particles were pre-incubated with the transduction enhancer human blood coagulation factor X (hFX) [23]. For both, transduction with HAdV-5 (combined with hFX) or HAdV-5-HexPos3 (without enhancer), efficiencies of ≥80% were achieved for BM-hMSCs and ≥50% for A-hMSCs in previous studies [23,29]. Six days post-infection, the supernatants were discarded, and remaining cells were fixed and stained with crystal violet (Figure 1). In this assay, high levels of crystal violet staining correspond to a high number of remaining cells indicating a low level of virus replication.

In the samples in which wildtype-capsid replication-competent HAdV-5 vectors were used for infection, no lysis of hMSCs was detected independent of pMOI or life cycle modification. In contrast, strong virus-mediated cytotoxicity was observed when HAdV-5 was combined with the transduction enhancer hFX or when HAdV-5-HexPos3 particles were used, independent of life cycle modification. Interestingly, while no differences were observed between vectors without life cycle modification and with E1A-Δ24 modification, infection of cells with E1A-Δ24-E1B-Δ19K vectors resulted in a different phenotype independent of the pMOI used (between 1 and 1000). Upon infection with replication-competent vectors without life cycle modification or with the E1A-Δ24 modification, hMSCs show a roundish phenotype and slowly detach from the culture plate. In contrast, upon infection with vectors carrying the E1A-Δ24-E1B-Δ19K life cycle modification, hMSCs seem to be destroyed by the virus infection without previously detaching from the culture plate (Figure 2). Notably, the cytotoxic effects of HAdV-5-HexPos3 particles were significantly stronger compared to HAdV-5 particles combined with hFX, which is why only HAdV-5-HexPos3 particles were used for all further experiments.

Next, we determined the amount of viral progeny produced at different time points by hMSCs infected with HAdV-5-HexPos3 with or without life cycle modification. Every 24 h, cells and supernatant were harvested, and the number of viral particles produced per hMSC was quantified by qPCR (Figure 3).

Generation of viral progeny from hMSCs was demonstrated in all samples, with no significant differences between the different replication-competent vectors or the pMOI used for infection. In BM-hMSCs, a maximum of about 1 × 10^4^ viral particles produced per cell was measured, whereas production efficiency in A-hMSCs was slightly lower. Independent of the hMSC origin, the maximum amount of viral progeny was already reached 48–72 h post-infection. Thus, infection of hMSCs with different replication-competent HAdV-5-HexPos3 vectors results in equally efficient viral replication.

### 3.2. Replication-Competent HAdV-5-HexPos3 Vectors Carrying the E1B-Δ19K Mutation Show Accelerated Replication and Release of Mature Viral Progeny in hMSCs

To confirm that viral progeny released from hMSCs are mature and infectious, we infected BM- and A-hMSCs with the different replication-competent HAdV-5-HexPos3 vectors (pMOI 900) and seeded them onto a dense layer of UD-SCC-2 cells. As all vectors used for infection carried an eGFP transgene, infected hMSCs were detected as single eGFP-expressing cells in the wells. This enabled the analysis of virus release kinetics: if only single cells are visible in the well, the hMSCs have not yet been lysed. The appearance of eGFP-positive plaques, indicates hMSC lysis and spread of viral progeny to tumor cells. To assess viral spread to the UD-SCC-2 cells, the number of single cells (=non-lysed hMSCs) and the number of eGFP-positive plaques were counted using a fluorescence microscope and set into proportion every 24 h. In addition, the plaque diameter was determined (Figure 4A–C).

The plaque assays revealed a highly accelerated release of viral progeny for hMSCs infected with HAdV-5-HexPos3-E1A-Δ24-E1B-Δ19K. Already 48 h post-infection, first plaques were visible, and most of the single cells were lysed 96 h post-infection. In contrast, in samples with hMSCs infected with HAdV-5-HexPos3 (no life cycle modification) or HAdV-5-HexPos3-E1A-Δ24, only single eGFP-expressing cells were detected until 72 h post-infection, and even after more than 200 h, single eGFP-expressing hMSCs were still visible. This difference in replication kinetics was also reflected in the determined plaque diameters. While the plaque diameter was already >1 mm at 216 h after infection of hMSCs with the E1B-Δ19K mutant, plaques induced by hMSCs infected with the two other viruses were still <0.5 mm in diameter. Results were comparable independent of hMSC origin (data not shown).

The accelerated lysis of hMSCs infected with HAdV-5-HexPos3-E1A-Δ24-E1B-Δ19K was additionally verified by a lactate dehydrogenase (LDH) assay (Figure 4D). Increased levels of LDH were detected in the cell culture supernatant of HAdV-5-HexPos3-E1A-Δ24-E1B-Δ19K-infected hMSCs from 48 h onwards compared to hMSCs infected with the other replication-competent vectors, corresponding to an accelerated release of viral progeny upon infection with E1B-Δ19K-modified vectors.

### 3.3. Migration of hMSCs Is Not Inhibited by Virus Infection In Vitro

Equally essential as efficient virus replication in hMSCs is their migration toward tumor cells. As virus infection results in substantial interference with cellular homeostasis, we aimed to investigate whether the migration behavior of hMSCs was altered upon virus infection. First, we analyzed the migration of non-infected BM- and A-hMSCs toward the conditioned medium of the HNSCC cell lines UD-SCC-2 and UD-SCC-6 using a Boyden chamber assay (Figure 5A,B). Human MSCs were attracted by the conditioned media, as significantly more cells had migrated toward the tumor cell-conditioned medium compared to the unconditioned control medium. No difference was observed between the conditioned medium of the two HNSCC cell lines. Next, we analyzed the migration of hMSCs infected with the HexPos3 variant of the different replication-competent HAdV-5 vectors. For both types of hMSCs, no statistically significant difference in migration toward UD-SCC-2 conditioned medium was detected when infected hMSCs were compared to their non-infected counterparts (Figure 5C,D).

### 3.4. No Migration of hMSCs toward a Xenograft HNSCC Tumor In Vivo

In order to evaluate hMSC migration toward HNSCC tumors in vivo, two different xenograft mouse models were established. To this end, 10 HNSCC cell lines were screened to determine how well they are transduced with HAdV-5-HexPos3 particles (Appendix A). Out of this portfolio, the HNSCC cell lines UD-SCC-2 and UD-SCC-6 were selected to establish a tumor model in NSG mice as these cell lines were efficiently transduced with both HAdV-5 and HAdV-5-HexPos3 vectors, showed suitable but not too fast cell growth in vitro, and represent the two major groups of HNSCCs: human papilloma virus (HPV)-positive (UD-SCC-2, tested positive for HPV-16) or -negative (UD-SCC-6) tumors (Appendix A) [1]. Upon subcutaneous injection of 2 × 10^6^ cells into the left flank of the NSG mice, we observed a 100% take rate, homogeneous tumor growth (Appendix A), and a high level of vascularization (Appendix A).

In order to analyze the migration of BM-hMSCs toward these xenograft tumors, cells were transduced with a replication-deficient HAdV-5 vector carrying a firefly luciferase gene (pMOI 1000). To enable efficient transduction of hMSCs (≥80% transduced cells), hFX was used as a transduction enhancer. Three hours post-transduction, BM-hMSCs were injected into the tail vein of tumor-bearing mice (14 or 21 days of tumor growth). Subsequently, firefly luciferase-expressing hMSCs (and a negative control having received non-transduced MSCs) were tracked using the IVIS 200 system. In animals injected with firefly luciferase-expressing BM-hMSCs, a decreasing luminescence signal was detected at the injection site and in the lungs, being strongest at 24 h post-hMSC injection (Figure 6A). Unexpectedly, no luminescence signal was detected at the tumor site at any time, although suitable vascularization of the tumors was confirmed by immuno-histochemical staining on the endothelial cell marker CD31 (Figure 7A,B).

As an alternative approach, we analyzed whether i.p.-injected BM-hMSCs migrate toward the subcutaneous xenograft tumors (injection after 21 days of tumor growth). Strong luminescence signals were detected in the abdomen and scrotum. (Figure 6B). Tumors of some i.v.- or i.p.-injected mice were removed 72 h after BM-hMSCs injection to analyze the luminescence of the isolated tumors directly. No luminescence signal was detected in the isolated tumors (Appendix A). Quantitative PCR analysis of tumor tissue was performed to determine intratumoral vector genome copy numbers. However, viral DNA was not detected in any of the samples (Figure 7C). In contrast, adenoviral DNA was detected in the lungs of mice injected with transduced hMSCs i.v. No differences were observed between the UD-SCC-2 and UD-SCC-6 tumor models. Due to the lack of hMSC migration to the xenograft tumors, we did not perform any long-term efficacy studies with hMSCs infected with replication-competent oncolytic adenovirus in this model.

### 3.5. No Migration of hMSCs toward a Xenograft HNSCC Tumor in a Chorioallantoic Membrane Model

In addition to the murine tumor models, we investigated the migration of BM-hMSCs in a recently published HNSCC CAM model [40]. For the engraftment of a tumor on the CAM, UM-SCC-11B cells were used. Four days after tumor engraftment, hMSCs (non-transduced or transduced with an eGFP-expressing HAdV-5-HexPos3 vector) and pure HAdV-5 vector were injected i.v. Additionally, transduced eGFP-expressing hMSCs were injected intratumorally. Delivery of pure HAdV-5 vector to the tumor upon i.v. injection was detected by qPCR and fluorescence microscopy 48 h post-injection (Figure 8). In addition, intratumorally injected hMSCs were detected by fluorescence microscopy in the xenograft tumor (Figure 8B). However, neither qPCR analysis nor fluorescence microscopy revealed any tumor infiltration by i.v.-injected hMSCs.

## 4. Discussion

Using hMSCs as carrier cells for oncolytic adenoviral vectors (oAVs) holds promise to enable systemic delivery and expands therapeutic options for the treatment of a range of cancer entities, including head and neck squamous cell carcinomas (HNSCCs). In this study, we investigated whether the recently published Hexon-modified vector HAdV-5-HexPos3 can serve as a tool for hMSC-based oncolytic therapy [29]. Moreover, we analyzed hMSC migration toward HNSCC xenografts in vivo and in ovo.

Investigation of different replication-competent HAdV-5-HexPos3 vectors revealed that increased infection efficiency of Hexon-modified particles led to efficient virus replication in hMSCs. Moreover, mature viral progeny are released and can re-infect HNSCC cell lines in vitro. This holds not only true for life cycle-unmodified HAdV-5-HexPos3 but also for the life cycle-modified conditionally replicating vectors HAdV-5-HexPos3-E1A-Δ24 and HAdV-5-HexPos3-E1A-Δ24-E1B-Δ19K. The E1B-Δ19K modification hereby highly accelerated the release of viral progeny. This is in line with previous reports demonstrating that E1B-Δ19K vectors are very potent oncolytic adenoviruses, showing increased tumor cell killing as well as accelerated virus release in tumor cells and hMSCs compared to life cycle-unmodified HAdV-5 [26,36,41,42,43,44]. However, the mechanisms leading to this accelerated virus release and subsequent spread in tumor tissue remain unclear. As the E1B 19K protein is a Bcl-2 analog functioning as an apoptosis inhibitor, increased apoptosis in HAdV-5-E1B-Δ19K-infected cells might explain accelerated virus release [45]. Moreover, E1B 19K is assumed to inhibit the proteolytic degradation of the nuclear lamins, a component of the nuclear membrane important for the stability of the nucleus [46,47]. The absence of E1B 19K could lead to proteolysis of the nuclear lamins, resulting in the disruption of the nuclear envelope and accelerated release of viral progeny. In addition, there is evidence that E1B 19K counteracts the adenovirus death protein (ADP), which is critical for the lysis of infected cells by adenoviruses [48]. Independently of the exact understanding of the mechanism, our results clearly demonstrate that the total number of viral particles produced per cell is not impeded. Although this might be an advantage concerning possible oncolytic effects, it is essential to consider that accelerated release of viral progeny also implies reduced time for carrier cells to migrate toward their target.

In our studies, we observed that non-infected and infected hMSCs efficiently migrated toward HNSCC cell lines in vitro. However, this tumor-homing capability was neither observed in two human HNSCC xenograft tumor mouse models nor in a human HNSCC xenograft tumor chorioallantoic membrane (CAM) model. In the mouse models, we detected most of the hMSCs in the lungs after i.v. injection, an observation reported by other researchers, too [49,50]. Nevertheless, successful delivery of oncolytic adenovirus using hMSCs as carrier cells was previously reported [21,25,50,51,52]. For example, Zielske et al. showed that hMSCs administered i.v. into the tail vein infiltrated the tumor tissue of subcutaneous colon cancer, breast cancer, and HNSCC xenografts in mice [52]. Interestingly, the lowest number of hMSCs was detected in the HNSCC xenografts, suggesting that hMSCs might be comparatively weakly attracted by this type of tumor cells [52]. During the establishment of our Boyden chamber migration assay, we also observed that BM-hMSCs migrated stronger to Huh7 or A549 cells compared to the HNSCC cell lines UD-SCC-2, UD-SCC-6, and UM-SCC-11B, supporting this hypothesis (data not shown). Reports of MSCs migrating to the tumor site have not only been published after i.v. but also after i.p. administration of the carrier cells [25,51,53]. For example, Moreno et al. showed high antitumor efficacy of menstrual blood-derived hMSCs loaded with oncolytic adenovirus upon i.p. injection in a lung adenocarcinoma xenograft tumor model in mice [51]. Thus, in contrast to our observations, several publications show that migration of MSCs toward (xenograft) tumors have been observed in murine models. Kaczorowski et al. reported migration of i.v.-injected hMSCs toward MiaPaCa-2 cells in the CAM model; however, we did not observe any migration of hMSCs to our xenograft tumors in ovo [54].

Apparently, the HNSCC cell lines used in our study did not sufficiently attract hMSCs in vivo and in ovo. Indeed, inefficient homing of hMSCs toward the target cells has often been described in the literature, and most studies show that only a limited proportion of hMSCs infiltrate the tumor tissue [49,50,52]. Human MSCs were shown to be capable of infiltrating a variety of tissues, including tumors, kidneys, lungs, liver, thymus, and skin. Nonetheless, achieving a more targeted migration of MSCs toward the tissue of interest is one of the major challenges in the use of hMSCs as carriers [55,56]. To reach this goal, researchers evaluate different strategies, such as increasing the chemokine secretion of tumor cells by radiation or improving the inherent migration ability of hMSCs by increasing the expression of proteins relevant for their homing such as the C-X-C chemokine receptor type 4 (CXCR-4) [52,57,58,59,60,61]. However, to guide hMSCs specifically and efficiently to a target tissue, a better understanding of the mechanisms underlying hMSC migration is mandatory and must be further investigated. Apart from the tumor cells, also the cells of the tumor microenvironment (TME), including endothelial cells, fat cells, and immune cells, are known to secrete chemoattracting molecules [62]. A combination of these molecules derived from the different cell types of the TME might be essential in the context of hMSC-based oncolytic virotherapy to efficiently attract hMSCs to the tumor. Since we used an immune-incompetent model, the lack of immune cell-associated chemoattracting molecules might be one possible explanation for the lack of hMSC homing, as these cells are known to secrete high amounts of cytokines and growth factors [63,64,65]. While immunocompetent (HNSCC) tumor models are available for several species (mice, hamsters, nonhuman primates…), and the isolation of MSCs from different species could be easily implemented, the use of such a model introduces a different problem: most organisms are not (completely) permissive for HAdV-5 infection. Thus, infection of MSCs from another species with HAdV-5 would not result in the release of viral progeny, disabling the evaluation of oncolytic effects. An intensive effort has been made to find a permissive animal model; however, to date, no really suitable animal model could be identified [66]. To sum up, this illustrates the complexity of preclinical hMSC-based oncolytic adenovirus therapy and highlights that the establishment of a more suitable in vivo model is indispensable but not yet realizable.

## 5. Conclusions

In summary, we show that the Hexon-modified HAdV-5-HexPos3 mutant vector is a powerful tool for the development of hMSC-based oncolytic virotherapy. Virus replication, the efficient spread of viral progeny to HNSCCs, and the maintained migration ability of infected hMSCs to HNSCCs were demonstrated in vitro. Moreover, our data highlight the need to gain a better understanding of mechanisms underlying the homing of hMSCs in vivo and shows that the suitability of immune-incompetent xenograft tumor models for the analysis of tumor homing of i.v.- or i.p.-administered hMSCs is questionable.

## 6. Patents

A patent application has been filed relating to this work by Ulm University (European patent application #19204420.4).

## Figures and Tables

**Figure 1 viruses-15-00218-f001:**
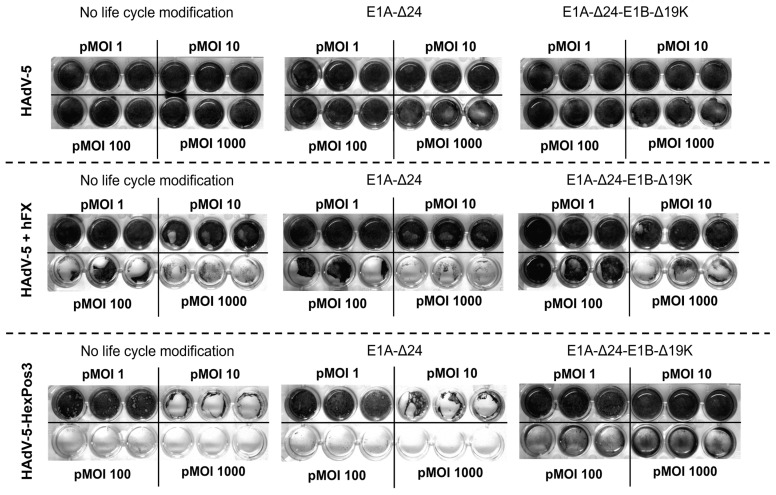
Replication-competent HAdV-5 carrying the HexPos3 mutation shows improved lysis of hMSCs. Human MSCs were infected with the indicated replicating virus (±human Factor X (hFX) as transduction enhancer) and indicated pMOIs. Six days post-infection, the wells were washed with PBS to remove all detached cells. Subsequently, remaining cells were fixed with 4% PFA and stained with crystal violet. After three more washing steps, plates were air-dried. The experiment was performed with BM- and A-hMSCs of three healthy donors. Representative images of one BM-hMSCs donor are shown.

**Figure 2 viruses-15-00218-f002:**
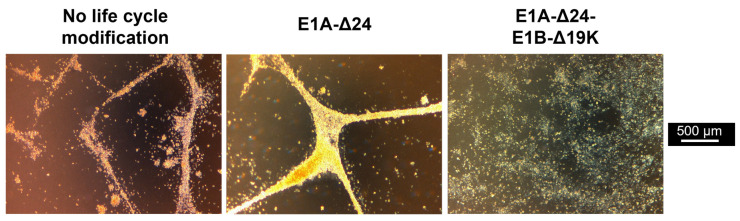
Infection of hMSCs with E1B-Δ19K-encoding virus results in a cytopathic effect with a different phenotype. BM-hMSCs were infected with pMOIs between 1 and 1000 using HAdV-5-HexPos3 particles harboring the indicated life cycle modification. Representative microscopic images of one hMSC donor, taken 72 h after infection with pMOI 1000, are shown.

**Figure 3 viruses-15-00218-f003:**
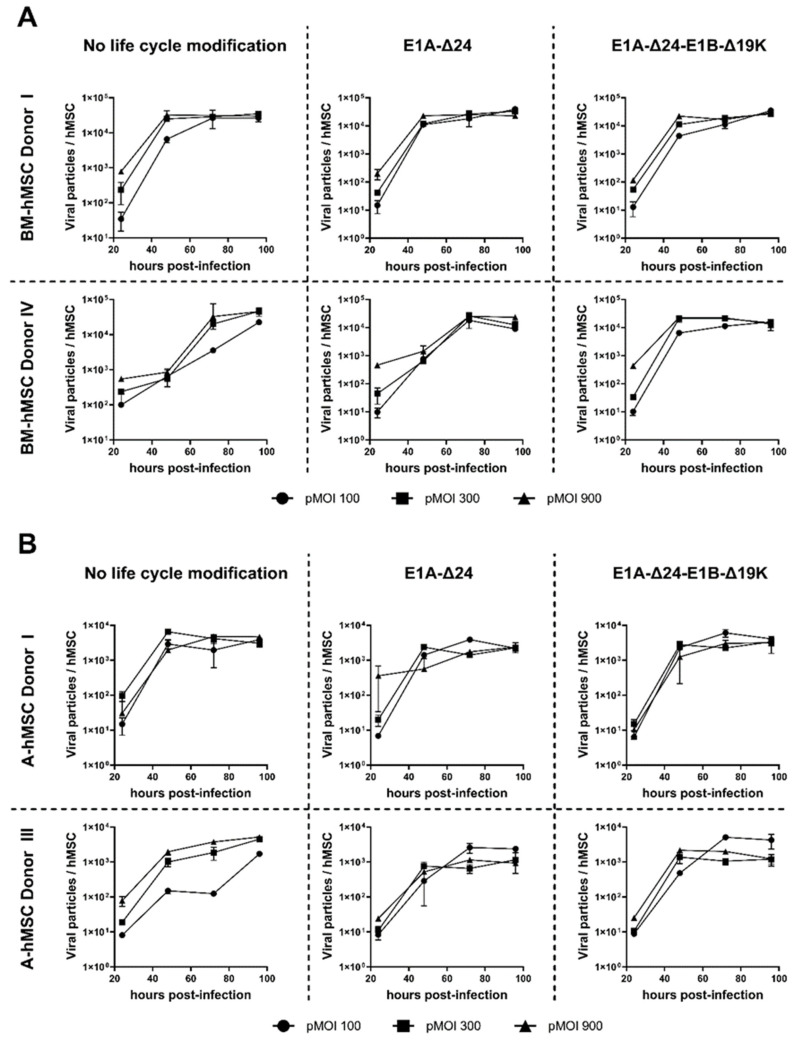
Replication of HAdV-5-HexPos3 in hMSCs is not affected by life cycle modifications. BM- (**A**) and A-hMSCs (**B**) were infected with the indicated replication-competent HAdV-5-HexPos3 vector at pMOI 100, 300, or 900. Every 24 h, cells and supernatant were harvested together to determine the amount of encapsidated adenoviral genomes by qPCR. Therefore, the harvested cells were disrupted in three freeze-thaw cycles and the lysate was treated with DNAse I to digest non-encapsidated viral genomes. Subsequently, samples were analyzed by qPCR to determine the number of adenoviral genomes using primers binding to the adenoviral E4 region. Results are shown as mean ± standard deviation of biological duplicates, each determined in technical triplicates.

**Figure 4 viruses-15-00218-f004:**
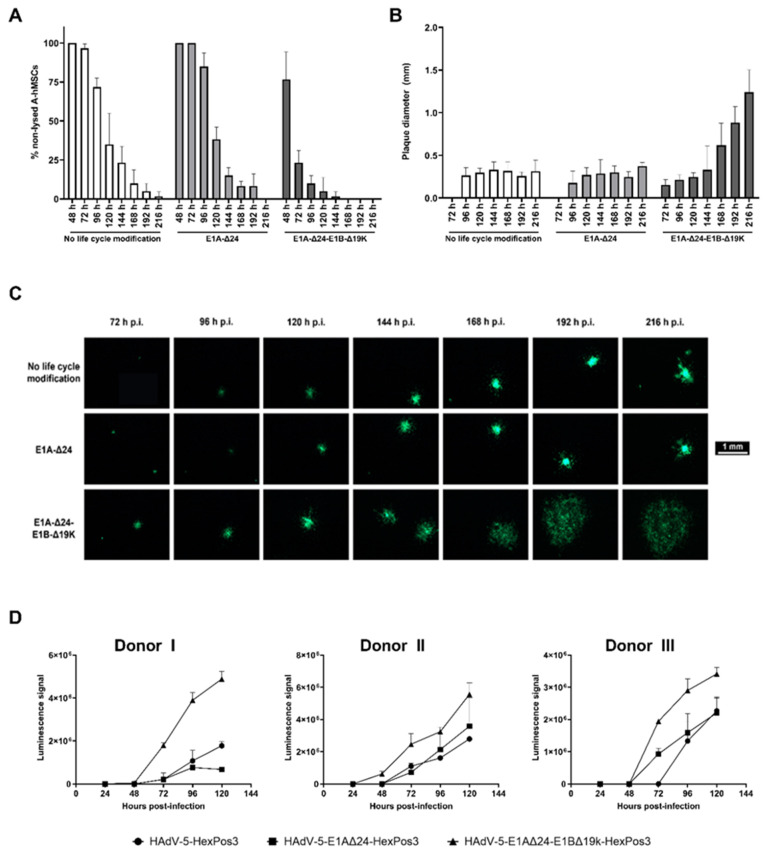
Release of viral progeny upon hMSC infection is significantly accelerated for replication-competent E1B-Δ19K-modified HAdV-5-HexPos3 vectors. (**A**,**B**) Human MSCs infected with an eGFP-expressing, replication-competent HAdV-5-HexPos3 vector with the indicated life cycle modification (pMOI 900) were co-cultured with UD-SCC-2 cells. Every day, the % of non-lysed hMSCs was determined by setting the number of single cells (=non-lysed cells) and the number of plaques visible under the fluorescence microscope in proportion ((**A**) mean ± standard deviation of biological triplicates). Moreover, the diameter of the occurring plaques was determined daily ((**B**) median ± interquartile ranges of biological triplicates). Representative results from one out of three A-hMSC donors are shown. (**C**) Example images of plaques in hMSC/UD-SCC-2 co-cultures. (**D**) hMSCs were infected with the indicated eGFP-expressing, replicating virus (pMOI 900). Starting after 24 h, the amount of LDH in the cell culture supernatant was determined using the LDH-Glo^®^ Cytotoxicity assay. Luminescence signals detected in samples of A-hMSCs infected with the indicated replication-competent HAdV-5-HexPos3 vector are shown as mean ± standard deviation of biological triplicates. All experiments were additionally performed with BM-hMSCs from three different donors leading to comparable results (data not shown).

**Figure 5 viruses-15-00218-f005:**
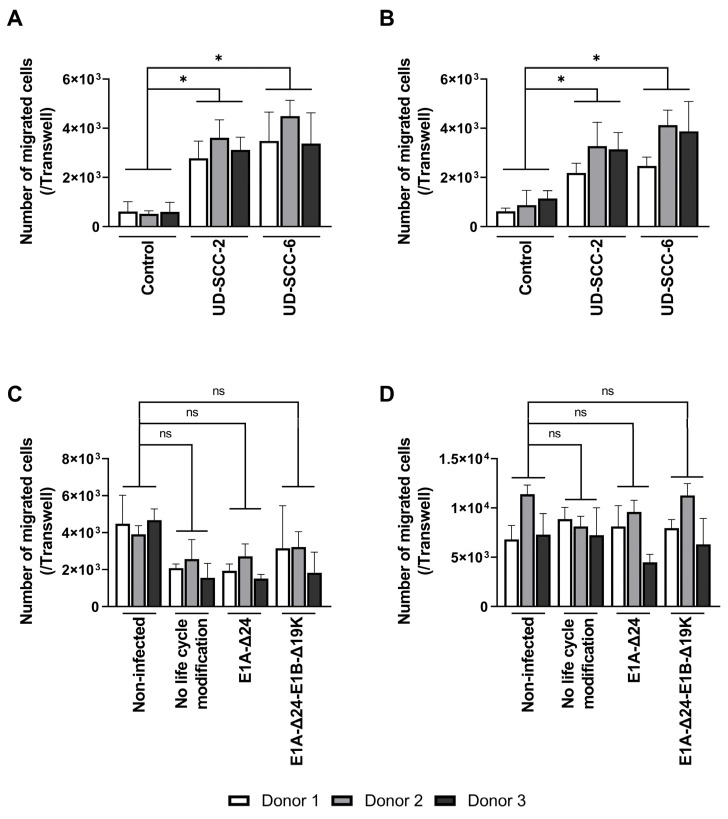
Migration of hMSCs toward HNSCC-conditioned medium is not inhibited by infection with replication-competent HAdV-5. (**A**,**B**) 2 × 10^4^ non-infected BM- (**A**) and A-hMSCs (**B**) were seeded into the upper well of a Boyden chamber, while the lower well contained unconditioned or cell line-conditioned medium. After 18 h, the total number of migrated hMSCs at the bottom side of the TC insert was determined. (**C**,**D**) BM- (**C**) and A-hMSCs (**D**) were infected with the indicated replication-competent HAdV-5-HexPos3 virus (pMOI 900) and transferred to the upper well of a Boyden chamber 3 h later. The lower well contained UD-SCC-2-conditioned medium. After 18 h, the total number of migrated hMSCs was determined. Results are given as mean ± standard deviation of biological triplicates. One-way ANOVA with subsequent Tukey’s multiple comparison was used for statistical analysis (* = *p* ≤ 0.05, ns = not significant).

**Figure 6 viruses-15-00218-f006:**
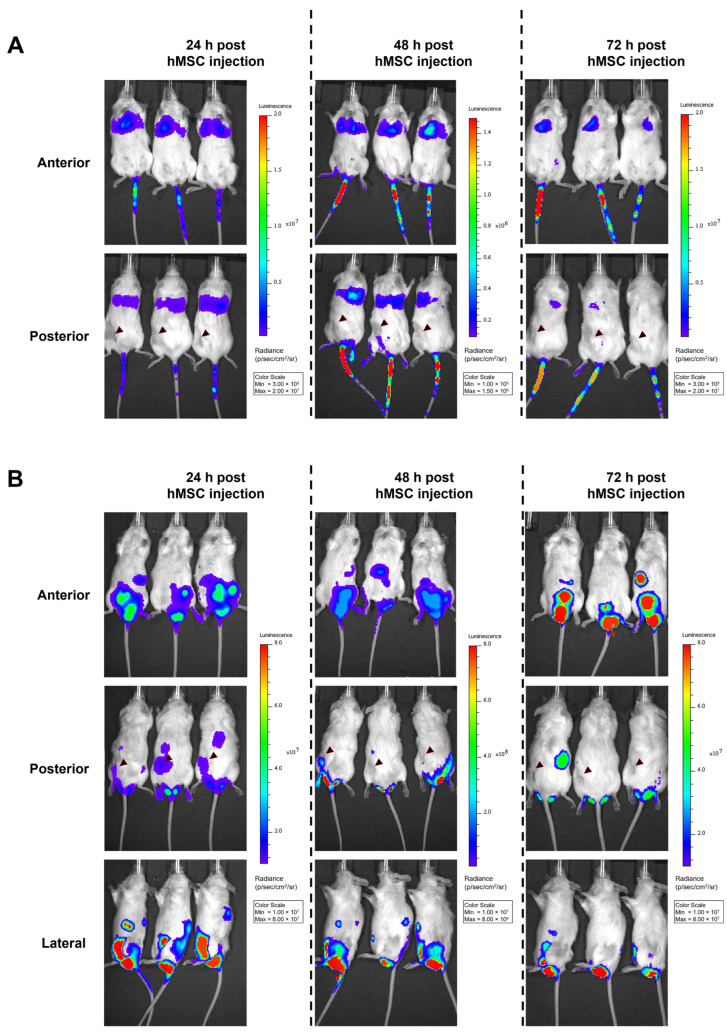
No migration of transduced hMSCs toward HNSCC xenograft tumors upon i.v. or i.p. injection in vivo. A total of 1 × 10^6^ BM-hMSCs (transduced 3 h before with a firefly luciferase-encoding replication-deficient HAdV-5 vector + hFX as transduction enhancer) were injected either (**A**) into the tail vein of tumor-bearing mice after 14 days of tumor growth or (**B**) i.p. after 21 days of tumor growth. Starting 24 h post-hMSC injection, luciferase signals were analyzed using the IVIS 200 in vivo imaging system. Five minutes before analysis, firefly luciferase substrate was injected into the mice i.p. Representative images from the IVIS 200 system are shown. The locations of xenograft tumors are indicated with a black arrowhead (posterior view). Representative images of UD-SCC-2 xenograft-bearing mice are shown. Similar results were obtained in UD-SCC-6 xenograft-bearing mice (data not shown). In total, transduced hMSC were injected i.v. into 18 mice (UD-SCC-2: *n* = 12 14 days post-tumor injection, *n*= 3 21 days post-tumor injection, UD-SCC-6: *n* = 3 21 days post-tumor injection) and i.p. (UD-SCC-2, *n* = 3 21 days post-tumor injection). Mock (non-transduced) hMSCs were injected into four mice as a control.

**Figure 7 viruses-15-00218-f007:**
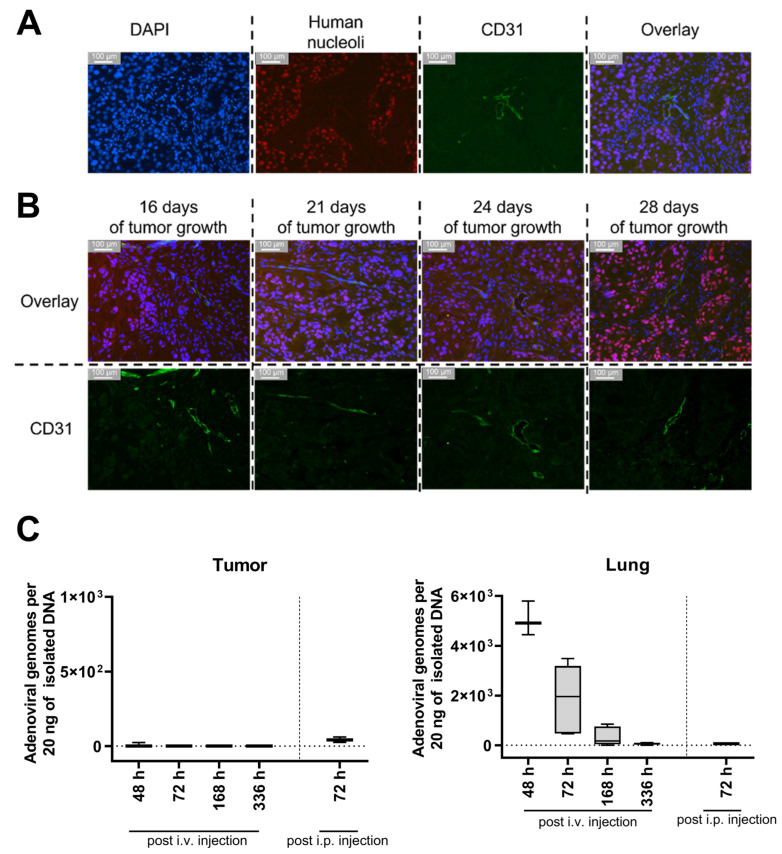
No adenoviral DNA was found in the xenograft tumor upon injection of BM-hMSCs transduced with a firefly luciferase-encoding HAdV-5 vector despite suitable tumor vascularization already 16 days after transplantation. (**A**,**B**) 6 µm tumor sections were stained using a rat α-mouse CD31 and a mouse α-human nuclear antigen antibody combined with two secondary antibodies (goat anti-rat IgG coupled to Alexa fluor 488, goat α-mouse coupled to Alexa fluor 594). Cell nuclei were stained with DAPI. (**A**) Staining of UD-SCC-2 xenograft tumor after 21 days of tumor growth. (**B**) Staining of UD-SCC-2 xenograft tumors after different times of tumor growth. Overlay = DAPI + Human Nucleoli + CD31 (**C**) Genomic DNA was isolated from snap-frozen UD-SCC-2, and UD-SCC-6 tumors and lungs from xenograft-bearing NSG mice having been injected with transduced BM-hMSC and analyzed for the presence of adenoviral E4 gene copy numbers by qPCR. Since the hMSCs were transduced with a replication-incompetent first-generation HAdV-5 vector before injection, the presence of E4 in the isolated total DNA can be considered an indicator for hMSC infiltration. At 48 h: *n* = 3; 72 h: *n* = 6; 168 h: *n* = 4; 336 h: *n* = 5; i.p.: *n* = 3. Data of UD-SCC-2 and UD-SCC-6 and both injection time points were taken together, as no differences were detected.

**Figure 8 viruses-15-00218-f008:**
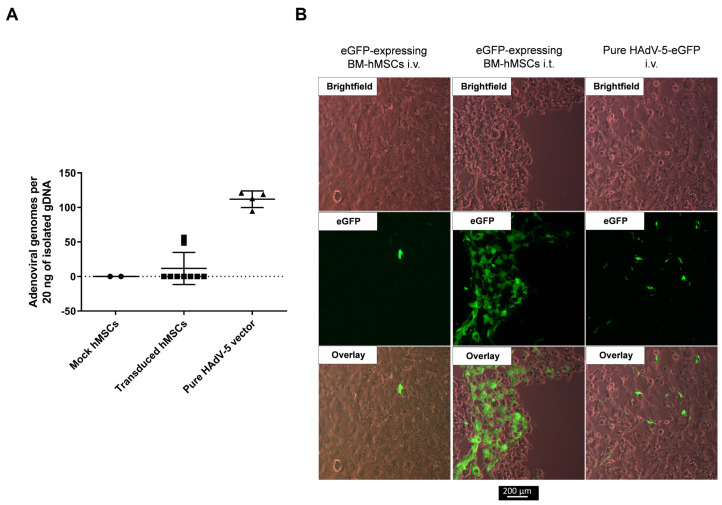
No infiltration of i.v.-injected hMSCs in UM-SCC-11B tumors in the CAM model. UM-SCC-11B xenograft tumor-bearing fertilized chicken eggs were injected either i.v. or i.t. with 5 × 10^5^ of either non-transduced hMSCs (mock) or hMSCs having been transduced with eGFP-expressing replication-incompetent HAdV-5-HexPos3 vectors 3 h prior to injection. Additionally, eggs were injected i.v. with 5 × 10^9^ pure eGFP-expressing replication-incompetent HAdV-5 vector particles. Injections were performed on day 4 after tumor cell application on the CAM. Tumors were isolated from the CAM 48 h post-vector/hMSCs injections and snap-frozen in liquid nitrogen. (**A**) Total DNA was isolated, and the copy number of the adenoviral E4 gene was analyzed by qPCR. Since hMSCs were transduced with a replication-incompetent first-generation HAdV-5 vector (expressing eGFP) before injection, the presence of adenoviral E4 in the isolated total tumor DNA can be considered an indicator for hMSC infiltration. Results of the qPCR analysis are shown as mean ± standard deviation (mock *n* = 2, transduced hMSCs *n* = 9, pure HAdV-5 *n* = 4). (**B**) Representative images of tumor sections analyzed by fluorescence microscopy.

## Data Availability

Data are contained and available within this manuscript.

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
