# Peer review of "Evaluation of Human Mesenchymal Stromal Cells as Carriers for the Delivery of Oncolytic HAdV-5 to Head and Neck Squamous Cell Carcinomas"

_viruses, 2023, doi:10.3390/v15010218_

Round 1

Reviewer 1 Report

This manuscript describes the use of mesenchymal stromal cells for delivery of adenoviral vectors to head and neck squamous cell carcinoma. The authors used a previously described adenovirus vector that contains four lysine residues to replace 13 mostly negatively charged amino acids from one of the hexon hyper variable loops. This adenovirus demonstrated to be effectively transducing human MSCs in culture. The adenovirus-transduced MSCs were used to deliver adenoviruses to cultured carcinoma cells in vitro, in vivo (in a murine xenograft model) as well as in ovo. In the latter models human tumors were grown on embryonated chicken eggs.

The data show that the vector system is efficient for infecting MSCs and that transduction does not impair the migration of MSC in a transwell-system model.

The manuscript is well written and the data shown generally suffice to support the conclusions drawn. There is, however a pressing issue needs attention.

The four lysine residues in het hexon protein form a positively charged pocket. The MSCs are cultured in medium with 500 U/mL heparin. The negatively charged heparin will neutralize the positive charges of the 4K lysine pocket once adenoviruses are added to the MSC medium. How does this affect the infection efficiency?  Does this affect the release of the virus from the infected MSC and the transduction of neighboring cells? Does the presence of heparin affect plaque size? These are very relevant issues as the adenovirus release and infection will have to take place in absence of heparin in the in vivo conditions.

Reviewer 2 Report

The manuscript deals with the topic of evaluating effectiveness of Hexon-modified HAdV-5-HexPos3 mutant vector as a powerful tool for the development of hMSC-based oncolytic virotherapy. In particular, the manuscript focuses on the analysis of virus replicate and release kinetics as well as the migration ability of infected hMSCs, three different types of Hexon-modified HAdV was generated. The manuscript demonstrated virus replication, efficient spread of viral progeny to HNSCCs, and migration ability of infected hMSCs to HNSCCs in vitro. In addition, the negative result of the homing of hMSCs to tumor lesion in vivo also emphasizes the importance to generate optimized in vivo models for the evaluation of hMSC as carrier cells.

General concept comments:

  1. The reason why oncolytic virotherapy using oncolytic adenoviral vectors (oAVs) is a promising approach to improve the treatment of HNSSCs need to be furtherly illustrated and cite some relating reference in introduction part.
  2. Total virus titers were determined by optical density measurement at 260 nm wavelength, whether the virus titers can be determined using the method of TCID50?
  3. Do infected hMSCs express adenoviral antigens as a result of replication ? If so, does this impact immunity to infected hMSC cells used? Does viral infection change the immune phenotype on MSCs? 
  4. Concerning in vivo studies, a question arises why tumor volume and mouse survival data was not documented. Is there significant difference between different groups?  
  5. The statistical analysis need to be supplement in Materials and Methods section.
  6. The method to measure the virus replication ability in BM-and A-hMSCs may be supplemented by the TCID50 and the fluorescence values of crystal violet  staining in Figure 1 need to be counted statistically.
  7. The tumor-infiltration situation of mock hMSCs (can be detected by IHC using MSC-specific antibodies) maybe need to supplement in Figure 8.

Specific comments:

1.      How to get the result :“maximum of about 1 × 104 viral particles produced per cell”in line 368,maybe the formula can be provided.

  1. The reason why authors selected UD-SCC-2 423 (HPV-16-positive) and UD-SCC-6 (HPV-negative) these two HNSCC cell lines to establish a tumor in NSG mice in line 423-425 need further explanation.
  2. The expression in line 435-437 is not very clear, and it needs to be described more accurately in combination with Figure 6A.

Reviewer 3 Report

1. For Figure 2, MOI 10 and 100 could also be explored.

2. For Figure 3B, A-hMSC Donor III, statistical analysis between groups can be performed.

3. OAd in hMSC and UD-SCC-2 could be detected after transduction or delivery.

4.  "Line 481-482, vector enables efficient replication of the virus in hMSCs" could be improved to "vector enables efficient  virus replication in hMSCs".

Round 2

Reviewer 1 Report

Thank you for clarifying het heparin issue. The changes to the manuscript remedy the issue. I can fully endorse the modified manuscript for publication.  

Reviewer 2 Report

The author has revised or clarified most of the questions adequately, however, question 3 maybe need further improve .Due to the infection of adenovirus, whether the immunogenicity of the hMSC cells has changed ? whether the phenotype of the hMSC cells  such as surface markers(CD90,CD73,CD105...) has changed? 
